# Heritage Designation and Urban Territorial Balance in Andalusia (Spain): An Approach towards Collaborative Methods in Rural Areas

**Blanca del Espino Hidalgo** 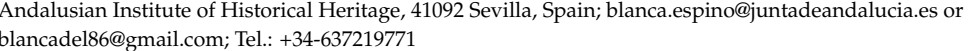

Andalusian Institute of Historical Heritage, 41092 Sevilla, Spain; blanca.espino@juntadeandalucia.es or blancadel86@gmail.com; Tel.: +34-637219771

**Abstract:** Numerous studies suggest that cultural heritage can be a powerful resource for local development when managed from the principles of sustainability and resilience. This paper aims to make a significant contribution to the designation of heritage assets. The case of the Andalusian region of southern Spain presents both qualitative and quantitative differences when a comparative study is made between urban centers, medium-sized cities, small towns, and rural areas. Subsequently, the paper proposes diverse methodologies to improve heritage designation in vulnerable territories through the incorporation of collaborative methods and digital humanities. The final objective is to conclude how to improve cultural heritage location and information processes to maximize social impact in areas suffering from aging and depopulation problems.

**Keywords:** urban territorial balance; heritage designation; collaborative mapping; documentation of cultural heritage; cultural heritage databases; territorial depopulation; local development; heritage-based development

## 1. Introduction

Within the European urban and social reality, rural areas have become a key subject of study, planning, and public policy. Currently, and despite the idea that we have of merely urban territory, according to official data [1], 137 million people in Europe live in rural areas. This represents almost 30% of the continent's population and more than 80% of its surface area. In cultural terms, the rural fabric is also responsible for many of the issues that characterize Europe: gastronomy and food production, natural, landscape and ecosystem diversity [2], or a large part of its traditions and rituals [3]. All these elements, which ultimately constitute the cultural heritage of these areas [4], contribute both to the legibility and transmission of European culture and identity and the generation of a positive impact on natural, social, and economic dimensions.

However, this cultural heritage is at risk due to the population movements that, in recent decades, have been affecting the socio-demographic reality of Europe and that, fundamentally, are reflected in the depopulation and aging of its rural areas [5]. According to the public consultation between 2020 and 2021 [6], rural and remote areas suffer from certain vulnerability factors that encourage their inhabitants to abandon the main urban centers. These factors include lack of basic services [7], lack or poor quality of mobility infrastructures, scarcity of employment opportunities [8], poor access to digital connectivity, and, in general, lack of participation or interest of rural society in decision-making processes [9]. However, the rural environment also presents certain advantages or opportunities, such as quality of life, preservation of the landscape character, sustainable agriculture, possibilities for social innovation, or an undeniable sense of belonging to the place [6].

In recent years, concern about population movements from rural areas to large metropolises has become a constant feature of the territorial policy, media impact, and

social crisis in much of Europe and the rest of the world. The south of the Iberian Peninsula is no stranger to this process, even though the collective imagination places regions such as Andalusia far from the territories marked by rural exodus, an attribute that is generally assumed in the center and north of Spain. On the contrary, the mountainous areas of inland Andalusia are marked by two trends: the loss and aging of the population.

Strategic territorial policy documents at the European level, such as the Atlas for the Territorial Agenda 2030 developed by the German Federal Institute for Building Research, Urban Affairs, and Spatial Development, reflect the demographic reality of this area at the municipal level. The data collection makes it possible to verify the presence of a strip that coincides with the entire Sierra Morena and others that coincide with the Sierra Bética, where the data are particularly alarming. In these areas, the average annual population growth, measured as a percentage gain or loss between the local censuses of 2001 and 2017, is negative in its entirety, alternating municipalities with declines of up to one point with others that lose between one and two or even more than two points [10] (p. 15).

On the other hand, in the Diagnosis of the National Strategy against the Demographic Challenge, in the axis aimed at aging, the conclusions reached are similar for this area. The map that allows us to visualize the rate of over-aging according to the Spanish municipal census of 2015 [11] (p. 18) distinguishes the aging data for the areas located to the south and the north of the coastal provinces.

Therefore, it is noted that the Andalusian territory is divided, from a demographic point of view, between a dynamic and young coastal strip, accompanied by the large urban centers and the Guadalquivir Valley, and a rural reality in the Sierras Morena and Béticas, where the data on depopulation and aging may well be assimilated to the areas already known in the center and north of Spain and Portugal.

A common characteristic of these territories is the tendency to abandon rural settlements [12]. Beyond the problem from a demographic and territorial planning point of view, this fact represents an added risk to the conservation of the rich cultural heritage they treasure [13]. Thus, in the rural areas of the Andalusian region, we find some of the most historically and territorially relevant archaeological sites [14], outstanding examples of religious architecture closely linked to traditions and rituals [15], and a multitude of architectural structures linked to the agricultural and livestock world [16], among other examples [17]. As for the vulnerability factors of these areas, they are shared with those we have already defined for European rural areas. Marked by the complicated communications between nuclei and heritage assets located at great distances between them, physical mobility is scarce. In demographic terms, the vast majority of municipalities suffer from aging and a sharp decline in population density [18].

Against this background, cultural heritage could be considered a potential factor for resilient territorial development in rural areas [19]. There have been many initiatives carried out in recent years in this regard. Most of them have certain factors in common, such as working with the local community [20], the use of digital networks to facilitate social interaction (particularly digital collaborative tools [21]), and the definition of elements of local identity [22]. Even so, grassroots initiatives are scarce although generally successful compared to those led by public administrations [23]. Thus, it seems logical that a better knowledge of the heritage assets in a certain area is necessary to increase the possibilities of local development based on the sustainable use of its cultural resources. This study will try to reveal to what extent the determination of cultural heritage assets in the southernmost region of Spain, Andalusia, is determined or not by the demographic dynamism of its different populations. In the second phase, the article will propose mechanisms for the improvement of the identification of heritage assets in the territories that most need it—rural areas and areas based on small towns—following the results achieved in the previous quantitative analysis.

According to official statistics [24], Europe has 19.3% of its population living in urban areas. The reality in Andalusia is quite different: 48.5% of its population lives in cities [25], which is much lower than the average European rural population of 29.1%. On the contrary,

the urban territorial structure in Andalusia is organized, for the most part, by networks of medium-sized cities, which constitute the territory inhabited by 38% of the region's population. Another figure that contrasts with the European average is the average population density, which in Andalusia is 40%, while on the continent it is 31.6%.

In general terms, it can be stated that the urban territorial structure of Andalusia is eminently urban and, in general terms, well-balanced. This is how it is defined by the Andalusian Spatial Plan (hereinafter, POTA), which establishes an analysis whereby, although there are small settlements and several rural areas in the region, most population centers are located at a relatively short distance from an urban center, whether it is a capital city, a main city or a medium-sized city. This fact has favored a certain stability in the demographic development of the entire region. However, in recent decades, this trend has been interrupted, with significant differences in population dynamics between areas that are growing significantly, mainly the large urban centers and coastal areas, and others that are progressively losing population, particularly the less populated and more dispersed urban structure.

Concerning their geographic characterization, most Andalusian rural areas are located in areas of high topography, particularly in the Betic and Sierra Morena mountain ranges. Moreover, they form part of traditional agrarian landscapes. In this sense, the loss of population has generated subsequent processes such as the reduction of the relevance of traditional agricultural production systems. This has generated great environmental risks such as the acceleration of soil erosion and desertification processes and, in the long run, a decrease in the cultural value of agrarian landscapes. Therefore, the distancing of the traditional relationship between the rural population of Andalusia, its environment and its territory [26] (p. 22), carries the risk of contributing to the loss of cultural values and practices that constitute important heritage assets in these areas.

The POTA structures the Andalusian rural territory based on three categories or types of spatial organization, which are related to their functional characteristics: networks organized by medium-sized cities, networks organized by rural centers, and other networks of rural settlements. In turn, the last category is divided into three others: dense networks of highly cohesive and homogeneous rural settlements, networks of rural settlements within centralized structures and, finally, networks of low-density rural settlements with weakly defined structures [26] (p. 29).

Finally, some remarks should be made about the categorization of medium-sized cities within the Andalusian urban territorial structure, which defines them qualitatively based on their intermediary role in the surrounding territory and not quantitatively according to their number of inhabitants. In fact, in population terms, they could be equivalent to what is known as small towns in other areas of Europe [27]. As we have already said, their great presence as a structuring element of the Andalusian territory makes them an asset for the population balance of the region. Particularly, some of the most stable at the population level are those in the interior of the region, which have been distanced from the dynamics of the provincial capitals and coastal areas. Known as agro-cities in the mid-twentieth century [28], they were considered settlements with the size of cities but with a socioeconomic structure typical of the agrarian world, lacking public services per their population size and linked to latifundia systems of land exploitation. Currently, they constitute authentic secondary urban centers, which supports them as a sample for this study.

## 2. Materials and Methods

First, a quantitative study will be developed to determine to what extent the demographic characterization of a territory or a typical human settlement determines the identification, documentation, and, therefore, designation of its cultural heritage. For this purpose, a series of data on all the municipalities of Andalusia, the southernmost region of Spain, has been taken as a reference. All data are taken from the Multi-territorial Information System of Andalusia (hereinafter, SIMA) [29], a multi-thematic and multi-territorial



statistical information database belonging to the Andalusian Institute of Statistics and Cartography (hereinafter, IECA). The data collected were as follows, all as of the last update in 2021.

- Population;
- Aging index;
- Immigration;
- Emigration;
- Number of immovable assets;
- Number of movable assets;
- Number of intangible assets;
- Surface area of the municipality in km$^2$.

From these primary data, secondary data have been created and used to obtain results and conclusions.

- Migratory balance, as the difference between immigration and emigration;
- Migratory balance per inhabitant, as the quotient between the migratory balance and the number of inhabitants;
- Population density, as the ratio between population and surface area;
- Real estate density, as the ratio between the number of real estate properties and the surface area per hundred;
- Density of movable assets, as the ratio between the number of movable assets and the surface area per hundred;
- Intangible asset density, as the ratio between the number of intangible assets and the surface area per hundred.

In sum, it has resulted in a set of 14 data taken for the 863 municipalities for which there are non-zero data in the registers for the year 2021 of the aforementioned information system. For processing, obtaining results, discussion, and drawing conclusions, the list of municipalities has been organized into four groups depending on their populations:

- 226 municipalities with less than 1000 inhabitants, which can be considered small settlements;
- 479 municipalities with between 1000 and 10,000 inhabitants, which can be considered small towns due to their functional characteristics;
- 145 municipalities with between 10,000 and 100,000 inhabitants, considered medium-sized cities according to their function and proportionality with the rest of the territory;
- 13 municipalities with more than 100,000 inhabitants, understood as main cities, which include the 8 provincial capitals in Andalusia and another 5 municipalities of great centrality in their surroundings.

The distribution has been carried out to compare the results obtained regarding the identification of heritage assets according to the population dynamics of each municipality. Likewise, it has been important to establish different intervals to obtain comparable results since they refer to assimilable urban entities. The population intervals for each category have been chosen according to the demographic characterization of the territory of Andalusia itself and the function within the urban territorial system of its urban and rural centers. It is likely that in other locations the definition of a medium-sized city, for example, would include cities with larger populations [30] (p. 10).

Given the nature of this publication, the sources used to update SIMA are varied: there are data from censuses, data obtained directly by the IECA or provided by other organizations, data from the exploitation of administrative records, etc. The concepts used are those defined by the source of origin [31].

Specifically, regarding data on heritage assets, these are collected by SIMA from those available in the Digital Guide to the Cultural Heritage of Andalusia [32], published by the Andalusian Institute of Historical Heritage (hereinafter, IAPH). It is a linked open data platform that provides access to all the information from the various databases on heritage

assets of this public institution through an integrated and intuitive query interface. In addition, through its incorporation into the Linked Open Data universe, its content is open to the re-use of data by third parties [33]. The Digital Guide to the Cultural Heritage of Andalusia contains information for the entire territory of Andalusia on:

- 27,987 assets of the immovable heritage;
- 98,774 movable heritage assets;
- 1841 intangible heritage activities;
- 117 cultural landscapes;
- 26 cultural routes.

These data are collected and updated annually by SIMA.

Complementary to the research and extraction of results from the quantitative data analysis, this work includes bibliographic and field research on the methods that are or could be used to improve the recognition and designation of heritage properties in areas characterized by a lower demographic dynamism. This method has mainly been applied in the final part of the article, namely in the Discussion section.

## 3. Results

The results obtained from the quantitative analysis of the data that have been systematized and calculated for all the municipalities of Andalusia, including demographic aspects, surface area, and cultural heritage assets, are presented below. First, a synthetic summary of the extracted data is presented (Table 1). Typical values of the most characteristic data have been obtained for each demographic category: small settlements, small towns, medium-sized cities, and main cities.

**Table 1.** Typical values * for each group of municipalities analyzed.

| Population Size | Number of Municipalities | Typical Population | Aging Index | Population Density | Built Heritage Density | Movable Heritage Density | Immaterial Heritage Density |
|---|---|---|---|---|---|---|---|
| Less than 1000 | 226 | 521 | 264 | 15 | 0.206 | 0.000 | 0.029 |
| 1000 to 10,000 | 479 | 3028 | 146 | 41 | 0.232 | 0.000 | 0.027 |
| 10,000 to 100,000 | 145 | 20,263 | 93 | 147 | 0.332 | 0.140 | 0.014 |
| More than 100,000 | 13 | 174,356 | 117 | 1.116 | 2.160 | 1.961 | 0.029 |

* To calculate the typical value, the statistical median has been used and not the arithmetic mean or average value for each municipality in each category. The statistical median is calculated as the central value of all the values obtained for the same data, ordered from smallest to largest. It is particularly suitable for a series of a large number of values as well as to avoid bias derived from outliers in the series. In descriptive statistics, the median is also called the "location parameter" and is used to express the central tendency of the data set.

We can observe how population density is, as might be expected, proportionally higher as the number of inhabitants of each municipality increases. The same does not occur with the aging index. It is found that the lowest value corresponds to the medium-sized cities, followed by the main cities, then the small towns, and, finally, the most aged category is that of the small settlements.

As regards the density of recognized heritage assets, it is proportional to the municipal demographic dimension when speaking of built heritage and movable heritage, which obtains zero values in small towns and small settlements. However, intangible heritage assets show a higher recognition for small settlements and major cities, followed very closely by small towns, and a substantially lower value than the rest in the case of medium-sized cities.

Next, for a more in-depth analysis, pairs of variables have been selected to detect the existence or not of correlations between them by representing all the values through point clouds.

### 3.1. Correlation between the Local Population and the Aging Index

First, the relationship between the local population, measured by the number of inhabitants of each municipality, and its aging index has been studied. After performing several tests with different groupings and categories of municipalities, the results of greatest interest have been obtained by discarding small settlements in rural areas as a population category and joining them with small towns (Figure 1). Medium-sized cities and major cities have been analyzed as independent categories (Figures 2 and 3).

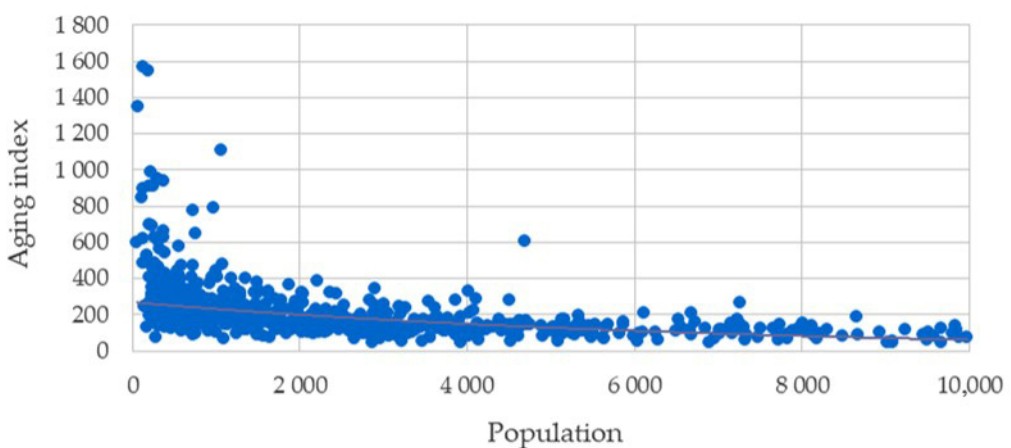

**Figure 1.** Correlation between the local population and the aging index. Rural areas and small towns.

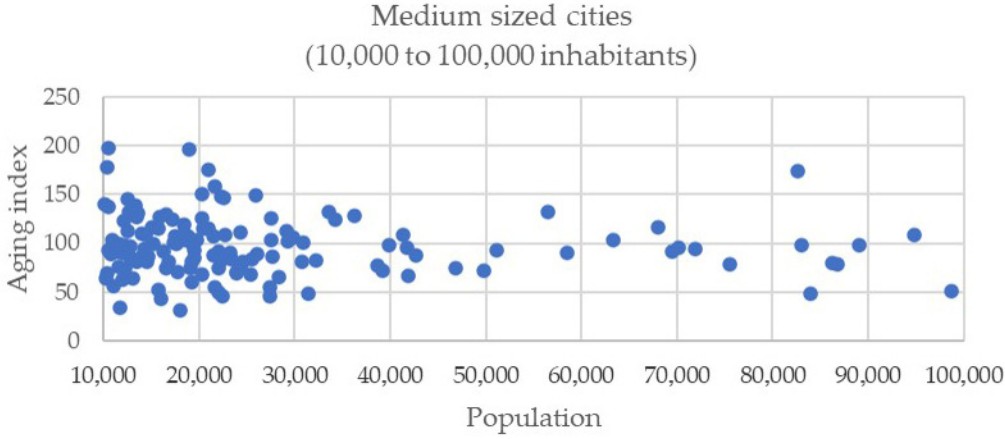

**Figure 2.** Correlation between the local population and the aging index. Medium-sized cities.

Figure 2 shows how the evolution of the aging index resembles an inverse hyperbolic function concerning the number of inhabitants of the municipality, with a very wide range of values. In contrast, the values are much more uniform in medium and large cities. The assignment of a correlation in these cases is also inconclusive. Cities with between 10,000 and 30,000 inhabitants present fewer uniform values (Figure 3), which approach a mean aging index of 100 as the population grows. This trend is maintained in the case of the main cities (Figure 4), with values very similar to the previous ones, which have been differentiated from them to avoid distortion on the horizontal axis of the graph.

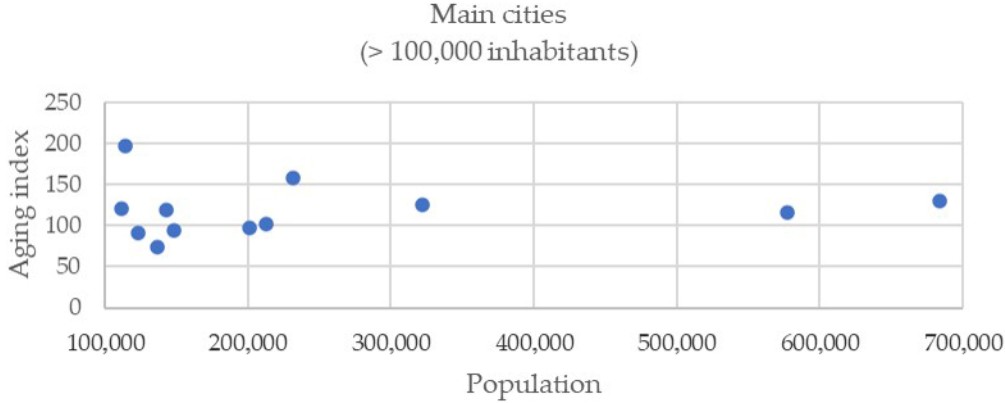

**Figure 3.** Correlation between the local population and the aging index. Major cities.

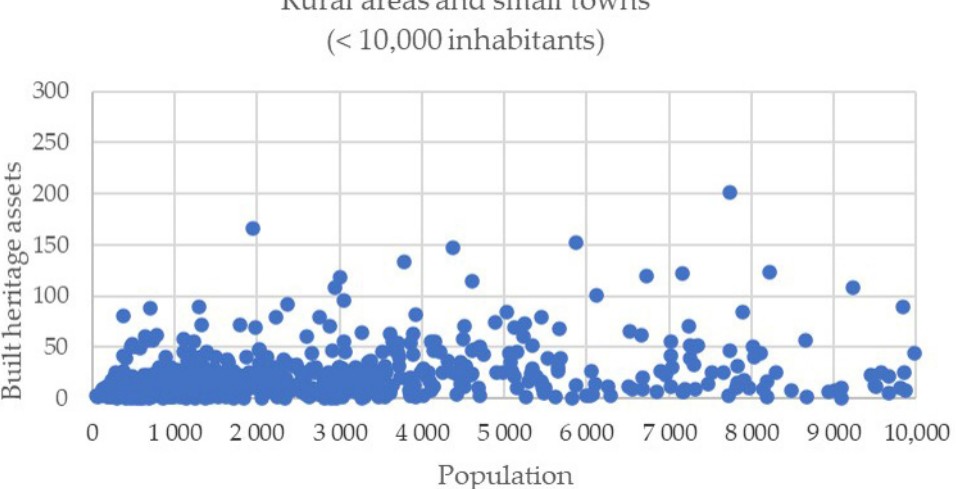

**Figure 4.** Correlation between the local population and the number of built heritage assets. Small settlements and towns.

### 3.2. Correlation between the Local Population and the Number of Built Heritage Assets

Next, the correlation between the number of inhabitants and the number of recognized cultural heritage properties will be analyzed. To do this, all the municipalities will be grouped into two categories, one consisting of small settlements and towns, and the other of medium-sized and major cities. The analysis will begin with the built heritage assets (Figures 4 and 5).

We can observe how the distribution of built heritage in the municipalities of smaller population size generally reaches data that are below 50 assets identified per municipality, with a second, less frequent level reaching 100 assets (Figure 4). In rare cases this figure is exceeded, reaching around 150 or, in a single case, reaching 200. In general terms, there is a greater concentration of low figures in the smaller municipalities, so that up to 3000 inhabitants none exceeds 100 identified assets. The data, however, do not show any correlation but tend to uniformity in the density of the point cloud between axes.

In the cities, however, the range of values is much wider (Figure 5). For medium-sized cities (less than 100,000 inhabitants), it is generally below 300 identified assets, although there are cases that reach 500 and 600. In major cities, the results are more diverse and generally wide, reaching 700 assets in one case. Although the dispersion of the graph is significant, which does not allow us to determine a correlation, there is a tendency to a parabolic function between the horizontal and vertical axes.

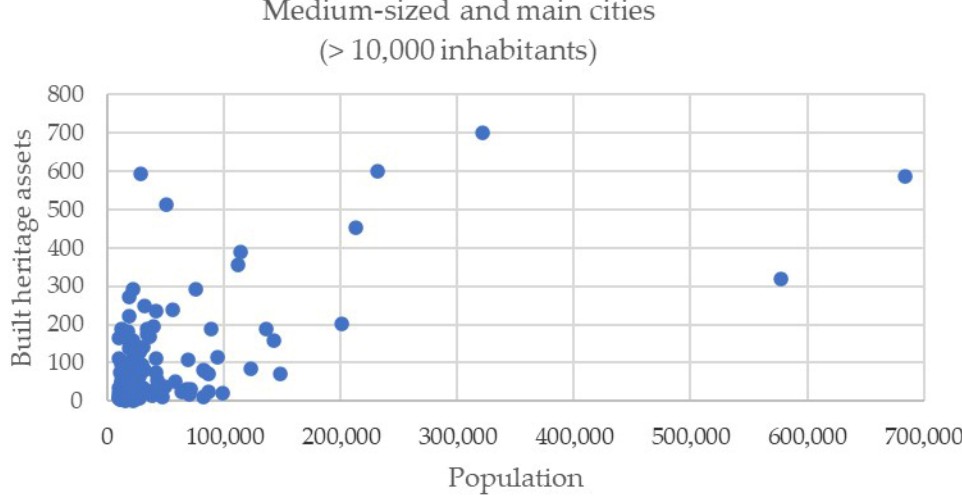

**Figure 5.** Correlation between the local population and the number of built heritage assets. Medium-sized and major cities.

*3.3. Correlation between the Local Population and the Number of Movable Heritage Assets*

The correlation between population and movable assets is analyzed below (Figures 6 and 7).

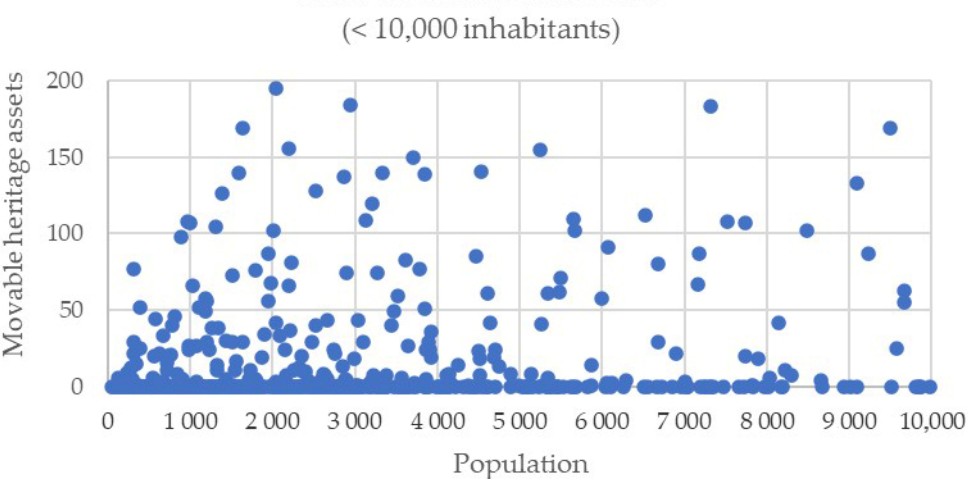

**Figure 6.** Correlation between the local population and the number of movable heritage assets. Small settlements and towns.

In the case of small settlements and towns (Figure 6), the analysis of the point cloud for each case yields inconclusive results. There is a large number of municipalities for which the results are null or practically null. The variable has a range that reaches, in the best cases, 150 and, very occasionally, almost 200 identified movable assets. There are no significant stages that can be related to population ranges, although it is clear that, in localities with less than 5000 inhabitants, the results are generally lower, with an abundance of 50 or fewer assets when the value is not null.

In medium-sized and major cities (Figure 7), there are also many municipalities with very low or practically no results. However, in contrast to the previous case, the range of values reached by the number of identified movable assets is much wider. In a good number of instances, it is around 2000 or 3000 assets, and there are cities in which figures of around 5000, 10,000, or even 20,000 are reached.

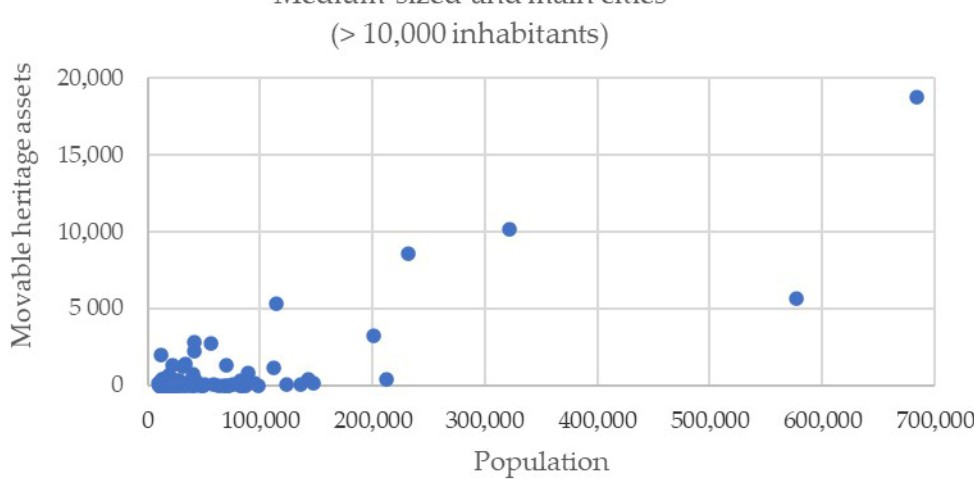

**Figure 7.** Correlation between the local population and the number of movable heritage assets. Medium-sized and major cities.

*3.4. Correlation between the Local Population and the Number of Immaterial Heritage Assets*

Finally, the possible correlation between the local population and the number of intangible heritage assets identified has been drawn (Figures 8 and 9).

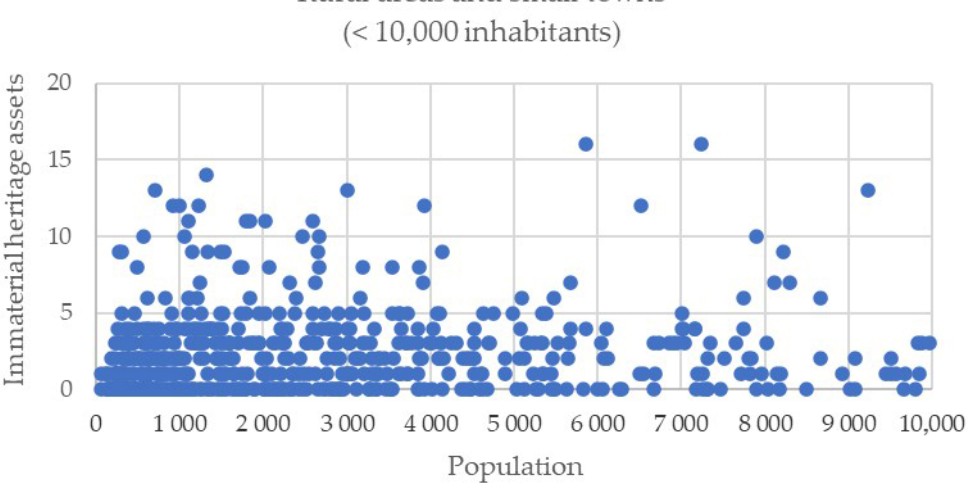

**Figure 8.** Correlation between the local population and the number of immaterial heritage assets. Small settlements and towns.

For settlements of smaller population sizes (Figure 8), the data for the identification of immaterial heritage properties show a clear pattern of around five properties per municipality. In general terms, the correlation does not exist, but rather the distribution is significantly uniform. Although there is an abundant number of municipalities with null data, it does not reach the proportion of the case of movable assets, but there is a wider range of values. There are cases of municipalities with 10 or even more than 15 identified immaterial heritage properties.

In the case of cities (Figure 9), there is also no certainty of a correlation between the number of inhabitants of the municipality and the number of immaterial heritage properties identified. In most cases, the records do not exceed 10 properties or, in the case of medium-sized cities, 15 per municipality, except in some major cities, where there are occasionally more than 20 or even 35 properties, which are not significant data from the recorded sample.

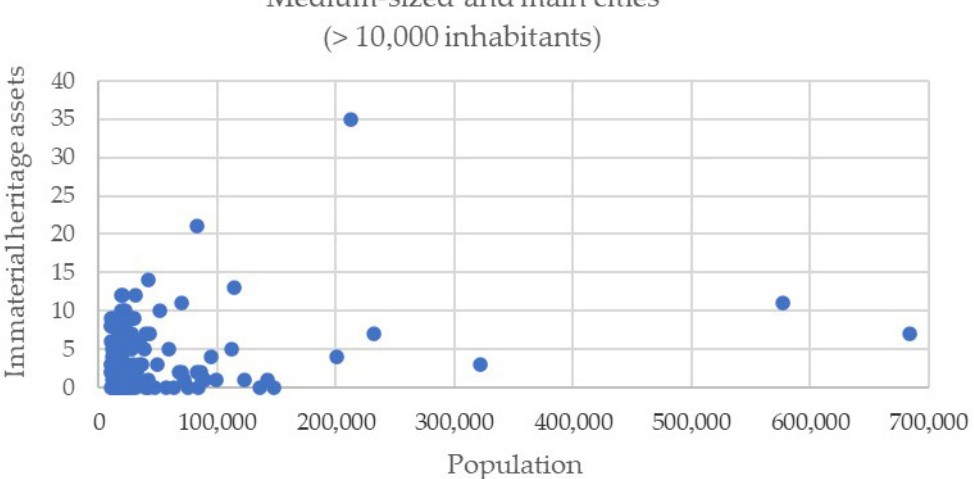

**Figure 9.** Correlation between the local population and the number of immaterial heritage assets. Medium-sized and major cities.

## 4. Discussion

The presentation of the results obtained from the analysis of demographic data, territorial distribution, and heritage assets reveals how the simplification in obtaining statistical indicators offers important clues for the establishment of conclusions. In this case, two fundamental facts stand out. On the one hand, the urban category with the least aging population is not that of large cities but, on the contrary, that of medium-sized cities, which supports the idea that this way of inhabiting the territory is ideal in terms of social balance and sustainability, as some authors have pointed out [34]. On the other hand, the relationship between the intensity of recognition of heritage assets and the population size of a municipality is not direct for all categories. It is true for immovable assets, while it is very disparate in the case of movable ones. However, when dealing with intangible heritage, it can be observed how the density of recognition of heritage activities in small settlements in rural areas is very similar to that of large cities, while the category with the lowest density is precisely that of the medium-sized Andalusian cities.

Furthermore, the consideration of each piece of data individually, but offered through their joint representation in graphs based on point clouds, offers the possibility of detecting anomalies and correspondences with a greater degree of detail. In this sense, it can be noticed how there is a tendency for the results obtained in the identification of movable and immovable heritage to shoot up as the demographic profile grows in medium-sized and major cities, but this does not occur for small cities and rural or dispersed settlements, where the results are much more uniform within the differences of each category and particularly consistent in the identification of rural intangible heritage.

As already mentioned, this work starts from the position that it is necessary to reenforce the identification of heritage assets in places with greater territorial and demographic vulnerability. The determination of this vulnerability, among multiple possible factors [35], has been linked here to the variable of population aging, which has been found to maintain an inverse logarithmic relationship with the number of inhabitants of each municipality. This is why, for the propositional part of this paper, it will be considered how to balance the identification or designation of heritage assets in all categories but with a special emphasis on small settlements and small towns.

Previously, to understand the results obtained, it was necessary to understand how the databases on heritage assets in their three categories—immovable, movable, and intangible—were generated, whose content was subsequently transferred to the aforementioned Digital Guide to the Cultural Heritage of Andalusia, and which have been collected by SICA for statistical purposes.

- For immovable assets, in most cases, there was a massive dumping of assets protected by heritage laws or registered in municipal catalogs, in which the archaeological heritage scattered in the territory abounds over the rest [36]. Focused documentation campaigns have also been carried out on specific heritage, such as that of contemporary architecture. Generally speaking, this determines more reasonable or expected proportions between the documentation of cultural heritage and the territorial demographic distribution. This does not detract from the fact that the identification of immovable heritage assets in rural areas, marked by small settlements and towns, should be strengthened.

- In the case of movable assets, the results are marked by the generalized way of documenting records on the movable heritage of ecclesiastical origin and property of religious institutions, with a long history of documentation promoted by the Spanish State and the Autonomous Communities. This results in very exhaustive registers of movable assets associated with very specific buildings or with both civil and ecclesiastical properties of great relevance. For this reason, there is an abundance of municipalities with no or very low value in all population categories, while there are specific cases with a high number of identified movable assets. However, the difference between the documentation of small settlements and towns and that of the movable heritage of medium and major cities is still significant, where not only are the median values significantly higher in the density of assets, but also the dispersion of points shows a much more nourished reality in terms of their documentation.

- In the category of intangible heritage, it should be mentioned that the great majority of identified assets come from the development, between the years 2008 and 2014, of the documentation project called Atlas of the Intangible Heritage of Andalusia [37], promoted and implemented by the Andalusian Institute of Historical Heritage. The methodology for obtaining data, through the creation of documentation teams distributed by counties throughout Andalusia, favors greater uniformity in obtaining results, which can be observed both in the dot scatter plots and in the synthetic results of the median density of assets by population category.

In this light, it seems logical to think that the methodology used for the documentation of intangible heritage is the one that, according to the results obtained, would best meet the objectives of balancing the identification of heritage assets in rural areas and small towns and would be the most appropriate for the identification of heritage assets in small towns. In just 6 years, 1800 expressions of the intangible heritage of Andalusia were collected from an anthropological perspective and using participatory methodologies [38].

Indeed, the development and implementation of participatory mechanisms applied to different fields of the heritage world—not only documentation but also intervention, restoration, and dissemination—have been, in recent decades, the subject of research and proposals from different cultural heritage research and management organizations. In the case of the IAPH, the Network of Cultural Heritage Reporting Agents, which originated precisely in the work of the Atlas of the Intangible Heritage of Andalusia, stands out for its recent creation and significant impact. Mention should also be made of projects such as the Open Heritage Laboratory, initiated within the Europeana Food and Drinks project framework, which was intended to promote an idea of open government based on institutional transparency, collaboration, and networking. Several attempts have been made to design participatory safeguarding plans directly related to these lines of work, together with the social agents that participated in the Atlas of the Intangible Heritage of Andalusia. This has been carried out thanks to a series of Intangible Heritage Seminars, the RedPesca project, and the project Methodological Guide for the Design of Special Plans for the Safeguarding of Intangible Cultural Heritage [39].

More recently, research projects have been implemented in which, specifically, innovative tools have been generated for the management of cultural heritage to make it a more open, transparent, and participatory concept in response to issues such as the demographic challenge or the objectives of sustainable development. This is the case of LAPat, which has

developed techniques for the collaborative safeguarding of diverse heritage through the concept of the open heritage laboratory, or SIN_PAR (Innovation System for the Heritage of Rural Andalusia), which has combined personal work with heritage agents with new technologies to improve, among other objectives, knowledge of the dispersed and remote heritage of vulnerable, peripheral, border, or depopulated areas of the Andalusian rural territory [40]. Both projects are based on four fundamental premises:

- The consideration of heritage assets as contributors to improving quality of life, favoring social cohesion and the resilience of the territories to which they belong, playing a decisive role in sustainable development and territorial balance.
- The awareness that social and technological innovation in cultural heritage must be linked to the generation of methodologies that improve the relationship between cultural heritage management institutions and citizens for the more effective safeguarding of the assets.
- The certainty that, beyond regulatory developments, the best guarantee for the safeguarding of cultural heritage is the collaboration between administrations, specialists, public and private agents, and citizens.
- The fact that use of open data, free licenses, and the reuse of information on cultural heritage is the best guarantee for generating feedback from the society that best knows the heritage of its territory.
- Among them, the SIN_PAR project has been specifically aimed at recognizing the role that cultural heritage can play in the cultural and economic revitalization of rural territories. Thus, it has investigated how the management of cultural heritage could improve the urban territorial balance in these areas. To this end, the digital humanities are a fundamental resource to enhance the development of these places as they have allowed the connectivity of people, territories, and resources, alleviating the physical isolation of these nuclei. The value of the institutional, business, and human cooperation through the digital sphere has gained special prominence after the context marked by the recent pandemic, which has dispelled many of the previous skeptical opinions. More specifically, the actions carried out have been as follows.
- The development of innovation workshops for cultural heritage with agents working in the territories chosen as case studies. In addition to contributing in a very significant way to improving the diagnosis and making proposals, they have served as links for the detailed knowledge of the heritage reality of the rural areas on which the project has focused.
- The opening of a digital bank of sustainable initiatives on the use of cultural heritage in vulnerable, remote, peripheral, or rural areas has made it possible to learn about realities that are physically and conceptually distant from the initial ones and has provided ideas for the improvement of the methodologies used.
- The creation of an interactive and collaborative map of the rural heritage of Andalusia in which a cartographic viewer with basic starting information has been integrated with the incorporation of the georeferenced database on rural study areas present in the IAPH. The tool has allowed anyone to provide information on immovable and intangible heritage assets not available in the existing records. In addition, it has opened the possibility of adding agents working with the cultural heritage of these places to the map, favoring the replicability of the initiative and facilitating networking beyond the objectives and time frame of the project itself.

In general terms, the proposals are based on overcoming a model centered solely on the capacity of public administrations to determine what is cultural heritage and what are the heritage assets of a territory. On the contrary, the idea is to incorporate the active participation of local citizens effectively in generation of information, without renouncing quality criteria such as review and validation by experts, technical discretion, or thematic and geographic balance, among other possibilities. This will benefit cultural heritage considerations as a resource for local and territorial development as well as the construction

of a corpus of collaborative methods for heritage designation that can be extrapolated to any urban or rural entity in an international context.

## 5. Conclusions

In recent decades, cultural heritage has undergone a process of re-signification in which it has been considered a resource for local and territorial development, especially related to concepts such as community resilience or sustainable development. Therefore, it can be understood that the presence and identification of cultural heritage assets should be a determining factor for the improvement of living conditions in a given place, something that is particularly necessary for vulnerable contexts from a socio-demographic point of view, i.e., rural, peripheral, and remote areas.

The analysis of the cultural assets present in an information system of the Spanish region of Andalusia reveals significant differences in the intentionality of the designation of elements in urban areas, medium-sized cities, small towns, and rural areas with small settlements. In general, places with smaller populations have a lower density of identification of heritage assets. These settlements are also those that show greater vulnerability in demographic terms, which has been determined by considering their aging index.

This trend is broken in the case of intangible heritage assets, which have a heritage density as high in settlements with fewer inhabitants as in the main cities. This has been related to the fact that their identification has been carried out later than that of immovable and movable assets and, therefore, has incorporated participatory and collaborative methodologies in which the local citizenry had a relevant role and was listened to by the responsible administrations.

Recent initiatives and research projects are investigating how people who have first-hand knowledge of the territory and the heritage assets associated with it can contribute actively and effectively to the processes of heritage de-designation. Fundamentally, these methods are based on collaborative strategies through personal and community dialogue but also based on the use of digital technologies.

Limitations regarding the uniformity of the data collected should be considered both in the interpretation of the results and in future reviews of this issue. In this regard, it should be recalled that the intensity of identification and documentation of heritage assets has shown inequalities not only between the different categories of human settlement but also within each urban scale. An extensive review of other heritage databases, for example, through the analysis of local urban protection catalogues, would be appropriate to complement this study.

Finally, the results obtained, and the theoretical and practical implications detected and presented in the discussion should prove useful in decision-making processes affecting the designation, identification, and documentation of cultural heritage in all types of settlements or urban networks, especially in rural, vulnerable, or peripheral areas. Similarly, the conclusions drawn could serve as a basis for improving other issues relating to both the physical and digital accessibility of the heritage resources of these places, where knowledge and appreciation of their cultural heritage should be a competitive advantage for local development, the quality of life of their inhabitants, and the anchoring of the population.

**Funding:** This research was funded by Plan Andaluz de Investigación, Desarrollo e Innovación 2020, Consejería de Transformación Económica, Industria, Conocimiento y Universidades de la Junta de Andalucía, Call 2020, grant number PY20_00298 as well as by Call CEIS 2020, grant number PYC20 RE 029 IAPH. No APC was charged for this research.

**Data Availability Statement:** The data presented in this study are openly available in Zenodo at https://doi.org/10.5281/zenodo.7772257 (accessed on 25 March 2023).

**Acknowledgments:** The author would like to thank the research teams of the SIN_PAR and SIT_PAR projects for their inspiration and ideas that, within the framework of these projects, have contributed to the ideation of this article. She would also like to thank the staff of the Andalusian Historical Heritage Institute, especially the Center for Documentation and Studies, and all the people who have

contributed to the creation of a coherent, integrated, and robust methodology for identifying and collecting information on the vast and diverse heritage assets of Andalusia, the results of which are now shown in the Digital Guide to the Cultural Heritage of Andalusia. Thanks also to those who have dedicated their time to transferring the keys of this work to the author.

**Conflicts of Interest:** The author declares no conflict of interest. The funders had no role in the design of the study; in the collection, analyses, or interpretation of data; in the writing of the manuscript.

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
