# Peer review of "Heritage Designation and Urban Territorial Balance in Andalusia (Spain): An Approach towards Collaborative Methods in Rural Areas"

_land, doi:10.3390/land12050974_

Round 1

Reviewer 1 Report

I would like to thank the author for an interesting research topic. I would like to point out some suggestions in order to improve the quality of the manuscript.

The abstract is satisfactory, but I suggest adding a sentence about the importance of the research. Of course, if the number of limiting words allows it. Reduce the number of keywords.

The introductory part contains all the necessary elements for understanding the goal, the essence of the research. I notice that the author uses the first person plural in the methodology, I suggest not to use the first person singular and plural. Also, I think that the part of the literature review should be expanded in the text and the references should be strengthened.

The introductory part contains all the necessary elements for understanding the goal, the essence of the research. I notice that the author uses the first person plural in the methodology, I suggest not to use the first person singular and plural. Also, I think that the part of the literature review should be expanded in the text and the references should be strengthened. The results are presented very clearly through tables and figures, which are very understandable. I am not suggesting changes in that part. The correlations shown by the author are very clear.

The discussion is extensive and sufficient, but I suggest that in the conclusion, the text about the limiting circumstances, as well as the theoretical and applied significance of the research and the future implications of the results, should be strengthened.

I suggest adding a reference:

Gajić, T., Đoković, F., Blešić, I., Petrović, M.D., Radovanović, M., Vukolić, D., Mandarić, M., Dašić, G., ; Syromiatnikova, J.A., Mićović, A. (2023). Pandemic Boosts Prospects for Recovery of Rural Tourism in Serbia. Land, 12(3), 624. DOI: 10.3390/land12030624.

Pandemic Boosts Prospects for Recovery of Rural Tourism in Serbia. Land 2023, 12, 624. https://doi.org/10.3390/ land12030624 

POSSIBLE PUBLICATION OF THE MANUSCRIPT AFTER CORRECTION

Author Response

The author would like to thank Reviewer 1 for such a detailed, beneficial, and positive report. All the comments and proposals have been taken into consideration and incorporated as far as possible.

The abstract has been modified to incorporate a sentence on the importance of the research without increasing the number of words.

The entire text has been revised to avoid the use of the first-person plural.

The literature review has been expanded in the text and the references have been strengthened. The reference suggested has been read, analyzed, and incorporated into the article.

In the conclusion, the text about the limiting circumstances, as well as the theoretical and applied significance of the research and the future implications of the results, have been strengthened.

Reviewer 2 Report

This article appears to be well organized and structured and uses an appropriate methodology to analyze the qualitative and quantitative differences in the designation of heritage assets through a comparative study between urban centers, medium-sized cities, small towns, and rural areas. I didn't notice any significant errors in the text, figures, or tables. However, some mistakes need to be corrected:

in Figure 1, correct the values of the aging index,

correct the numbering of chapters 3.3 and 3.4.

Author Response

The author would like to thank Reviewer 2 for a beneficial and positive report. The errors detected have been corrected, whether in the figures (Figure 1), in the text (full revision), or in the headings numbering (3.3. and 3.4.4).

Reviewer 3 Report

the study is very significant since the final objective is to draw 14 conclusions on how to improve cultural heritage location and information processes to maximize 15 social impact in areas suffering from aging and depopulation problems. the heritage preservation is quite an large issue for any country, and i recommend this study published in Land. 

Author Response

The author would like to thank Reviewer 3 for such a positive report on the interest and impact of the research 

Round 2

Reviewer 1 Report

All changes have been made. The manuscript is proposed for publication

Reviewer 2 Report

In this version of the article, the expected limitations regarding the uniformity of the collected data of the analyzed area were clarified and guidelines for future research were given.